# Transformer in Transformer

**Kai Han**[1,2]  **An Xiao**[2]  **Enhua Wu**[1,3*]  **Jianyuan Guo**[2]  **Chunjing Xu**[2]  **Yunhe Wang**[2*]

[1]State Key Lab of Computer Science, ISCAS & UCAS
[2]Huawei Noah's Ark Lab
[3]University of Macau
{hankai,weh}@ios.ac.cn, yunhe.wang@huawei.com

## Abstract

Transformer is a new kind of neural architecture which encodes the input data as powerful features via the attention mechanism. Basically, the visual transformers first divide the input images into several local patches and then calculate both representations and their relationship. Since natural images are of high complexity with abundant detail and color information, the granularity of the patch dividing is not fine enough for excavating features of objects in different scales and locations. In this paper, we point out that the attention inside these local patches are also essential for building visual transformers with high performance and we explore a new architecture, namely, Transformer iN Transformer (TNT). Specifically, we regard the local patches (*e.g.*, 16×16) as "visual sentences" and present to further divide them into smaller patches (*e.g.*, 4×4) as "visual words". The attention of each word will be calculated with other words in the given visual sentence with negligible computational costs. Features of both words and sentences will be aggregated to enhance the representation ability. Experiments on several benchmarks demonstrate the effectiveness of the proposed TNT architecture, *e.g.*, we achieve an 81.5% top-1 accuracy on the ImageNet, which is about 1.7% higher than that of the state-of-the-art visual transformer with similar computational cost. The PyTorch code is available at `https://github.com/huawei-noah/CV-Backbones`, and the MindSpore code is available at `https://gitee.com/mindspore/models/tree/master/research/cv/TNT`.

## 1   Introduction

In the past decade, the mainstream deep neural architectures used in the computer vision (CV) are mainly established on convolutional neural networks (CNNs) [17, 12, 11]. Differently, transformer is a type of neural network mainly based on self-attention mechanism [35], which can provide the relationships between different features. Transformer is widely used in the field of natural language processing (NLP), *e.g.*, the famous BERT [8] and GPT-3 [2] models. The power of these transformer models inspires the whole community to investigate the use of transformer for visual tasks.

To utilize the transformer architectures for conducting visual tasks, a number of researchers have explored for representing the sequence information from different data. For example, Wang *et al.* explore self-attention mechanism in non-local networks [37] for capturing long-range dependencies in video and image recognition. Carion *et al.* present DETR [3], which treats object detection as a direct set prediction problem and solve it using a transformer encoder-decoder architecture. Chen *et al.* propose the iGPT [5], which is the pioneering work applying pure transformer model (*i.e.*, without convolution) on image recognition by self-supervised pre-training.

---

*Corresponding author.

35th Conference on Neural Information Processing Systems (NeurIPS 2021).

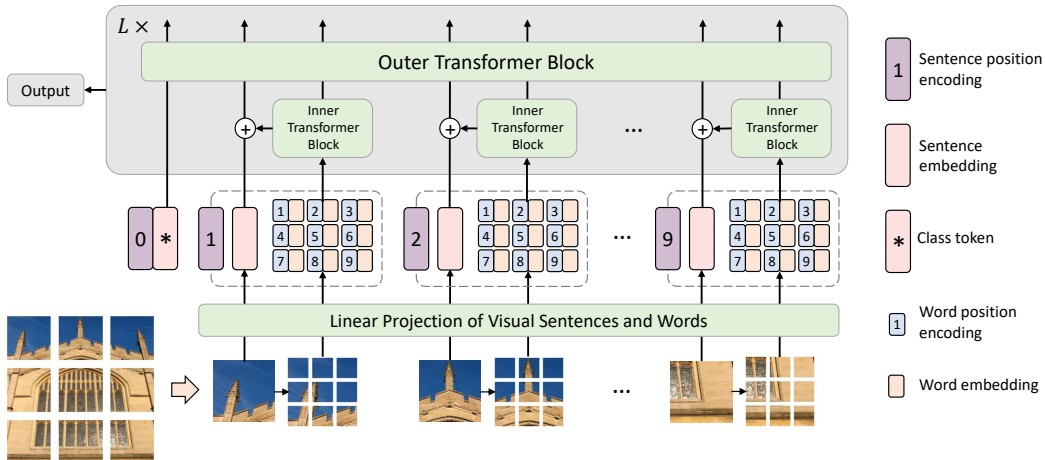

Figure 1: Illustration of the proposed Transformer-iN-Transformer (TNT) framework. The inner transformer block is shared in the same layer. The word position encodings are shared across visual sentences.

Different from the data in NLP tasks, there exists a semantic gap between input images and the ground-truth labels in CV tasks. To this end, Dosovitskiy *et al.* develop the ViT [9], which paves the way for transferring the success of transformer based NLP models. Concretely, ViT divides the given image into several local patches as a visual sequence. Then, the attention can be naturally calculated between any two image patches for generating effective feature representations for the recognition task. Subsequently, Touvron *et al.* explore the data-efficient training and distillation to enhance the performance of ViT on the ImageNet benchmark and obtain an about 81.8% ImageNet top-1 accuracy, which is comparable to that of the state-of-the-art convolutional networks. Chen *et al.* further treat the image processing tasks (*e.g.*, denosing and super-resolution) as a series of translations and develop the IPT model for handling multiple low-level computer vision problems [4]. Nowadays, transformer architectures have been used in a growing number of computer vision tasks [10] such as image recognition [6, 40, 29], object detection [46], and segmentation [43, 38].

Although the aforementioned visual transformers have made great efforts to boost the models' performances, most of existing works follow the conventional representation scheme used in ViT, *i.e.*, dividing the input images into patches. Such a exquisite paradigm can effectively capture the visual sequential information and estimate the attention between different image patches. However, the diversity of natural images in modern benchmarks is very high, *e.g.*, there are over 120 M images with 1000 different categories in the ImageNet dataset [26]. As shown in Figure 1, representing the given image into local patches can help us to find the relationship and similarity between them. However, there are also some sub-patches inside them with high similarity. Therefore, we are motivated to explore a more exquisite visual image dividing method for generating visual sequences and improve the performance.

In this paper, we propose a novel Transformer-iN-Transformer (TNT) architecture for visual recognition as shown in Figure 1. To enhance the feature representation ability of visual transformers, we first divide the input images into several patches as "visual sentences" and then further divide them into sub-patches as "visual words". Besides the conventional transformer blocks for extracting features and attentions of visual sentences, we further embed a sub-transformer into the architecture for excavating the features and details of smaller visual words. Specifically, features and attentions between visual words in each visual sentence are calculated independently using a shared network so that the increased amount of parameters and FLOPs (floating-point operations) is negligible. Then, features of words will be aggregated into the corresponding visual sentence. The class token is also used for the subsequent visual recognition task via a fully-connected head. Through the proposed TNT model, we can extract visual information with fine granularity and provide features with more details. We then conduct a series of experiments on the ImageNet benchmark and downstream tasks to demonstrate its superiority and thoroughly analyze the impact of the size for dividing visual words. The results show that our TNT can achieve better accuracy and FLOPs trade-off over the state-of-the-art transformer networks.

# 2 Approach

In this section, we describe the proposed transformer-in-transformer architecture and analyze the computation and parameter complexity in details.

## 2.1 Preliminaries

We first briefly describe the basic components in transformer [35], including MSA (Multi-head Self-Attention), MLP (Multi-Layer Perceptron) and LN (Layer Normalization).

**MSA.** In the self-attention module, the inputs $X \in \mathbb{R}^{n \times d}$ are linearly transformed to three parts, *i.e.*, queries $Q \in \mathbb{R}^{n \times d_k}$, keys $K \in \mathbb{R}^{n \times d_k}$ and values $V \in \mathbb{R}^{n \times d_v}$ where $n$ is the sequence length, $d$, $d_k$, $d_v$ are the dimensions of inputs, queries (keys) and values, respectively. The scaled dot-product attention is applied on $Q, K, V$:

$$Attention(Q, K, V) = softmax(\frac{QK^T}{\sqrt{d_k}})V. \tag{1}$$

Finally, a linear layer is used to produce the output. Multi-head self-attention splits the queries, keys and values to $h$ parts and perform the attention function in parallel, and then the output values of each head are concatenated and linearly projected to form the final output.

**MLP.** The MLP is applied between self-attention layers for feature transformation and non-linearity:

$$MLP(X) = FC(\sigma(FC(X))), \quad FC(X) = XW + b, \tag{2}$$

where $W$ and $b$ are the weight and bias term of fully-connected layer respectively, and $\sigma(\cdot)$ is the activation function such as GELU [13].

**LN.** Layer normalization [1] is a key part in transformer for stable training and faster convergence. LN is applied over each sample $x \in \mathbb{R}^d$ as follows:

$$LN(x) = \frac{x - \mu}{\delta} \circ \gamma + \beta \tag{3}$$

where $\mu \in \mathbb{R}$, $\delta \in \mathbb{R}$ are the mean and standard deviation of the feature respectively, $\circ$ is the element-wise dot, and $\gamma \in \mathbb{R}^d$, $\beta \in \mathbb{R}^d$ are learnable affine transform parameters.

## 2.2 Transformer in Transformer

Given a 2D image, we uniformly split it into $n$ patches $\mathcal{X} = [X^1, X^2, \cdots, X^n] \in \mathbb{R}^{n \times p \times p \times 3}$, where $(p, p)$ is the resolution of each image patch. ViT [9] just utilizes a standard transformer to process the sequence of patches which corrupts the local structure of a patch, as shown in Fig. 1(a). Instead, we propose Transformer-iN-Transformer (TNT) architecture to learn both global and local information in an image. In TNT, we view the patches as visual sentences that represent the image. Each patch is further divided into $m$ sub-patches, *i.e.*, a visual sentence is composed of a sequence of visual words:

$$X^i \rightarrow [x^{i,1}, x^{i,2}, \cdots, x^{i,m}], \tag{4}$$

where $x^{i,j} \in \mathbb{R}^{s \times s \times 3}$ is the $j$-th visual word of the $i$-th visual sentence, $(s, s)$ is the spatial size of sub-patches, and $j = 1, 2, \cdots, m$. With a linear projection, we transform the visual words into a sequence of word embeddings:

$$Y^i = [y^{i,1}, y^{i,2}, \cdots, y^{i,m}], \quad y^{i,j} = FC(Vec(x^{i,j})), \tag{5}$$

where $y^{i,j} \in \mathbb{R}^c$ is the $j$-th word embedding, $c$ is the dimension of word embedding, and $Vec(\cdot)$ is the vectorization operation.

In TNT, we have two data flows in which one flow operates across the visual sentences and the other processes the visual words inside each sentence. For the word embeddings, we utilize a transformer block to explore the relation between visual words:

$$Y'^i_l = Y^i_{l-1} + MSA(LN(Y^i_{l-1})), \tag{6}$$

$$Y^i_l = Y'^i_l + MLP(LN(Y'^i_l)). \tag{7}$$

where $l = 1, 2, \cdots, L$ is the index of the $l$-th block, and $L$ is the total number of stacked blocks. The input of the first block $Y_0^i$ is just $Y^i$ in Eq. 5. All word embeddings in the image after transformation are $\mathcal{Y}_l = [Y_l^1, Y_l^2, \cdots, Y_l^n]$. This can be viewed as an inner transformer block, denoted as $T_{in}$. This process builds the relationships among visual words by computing interactions between any two visual words. For example, in a patch of human face, a word corresponding to the eye is more related to other words of eyes while interacts less with forehead part.

For the sentence level, we create the sentence embedding memories to store the sequence of sentence-level representations: $\mathcal{Z}_0 = [Z_{\text{class}}, Z_0^1, Z_0^2, \cdots, Z_0^n] \in \mathbb{R}^{(n+1) \times d}$ where $Z_{\text{class}}$ is the class token similar to ViT [9], and all of them are initialized as zero. In each layer, the sequence of word embeddings are transformed into the domain of sentence embedding by linear projection and added into the sentence embedding:

$$Z_{l-1}^i = Z_{l-1}^i + FC(Vec(Y_l^i)), \tag{8}$$

where $Z_{l-1}^i \in \mathbb{R}^d$ and the fully-connected layer $FC$ makes the dimension match for addition. With the above addition operation, the representation of sentence embedding is augmented by the word-level features. We use the standard transformer block for transforming the sentence embeddings:

$$\mathcal{Z}'_l = \mathcal{Z}_{l-1} + MSA(LN(\mathcal{Z}_{l-1})), \tag{9}$$
$$\mathcal{Z}_l = \mathcal{Z}'_l + MLP(LN(\mathcal{Z}'_l)). \tag{10}$$

This outer transformer block $T_{out}$ is used for modeling relationships among sentence embeddings.

In summary, the inputs and outputs of the TNT block include the visual word embeddings and sentence embeddings as shown in Fig. 1(b), so the TNT can be formulated as

$$\mathcal{Y}_l, \mathcal{Z}_l = TNT(\mathcal{Y}_{l-1}, \mathcal{Z}_{l-1}). \tag{11}$$

In our TNT block, the inner transformer block is used to model the relationship between visual words for local feature extraction, and the outer transformer block captures the intrinsic information from the sequence of sentences. By stacking the TNT blocks for $L$ times, we build the transformer-in-transformer network. Finally, the classification token serves as the image representation and a fully-connected layer is applied for classification.

**Position encoding.** Spatial information is an important factor in image recognition. For sentence embeddings and word embeddings, we both add the corresponding position encodings to retain spatial information as shown in Fig. 1. The standard learnable 1D position encodings are utilized here. Specifically, each sentence is assigned with a position encodings:

$$\mathcal{Z}_0 \leftarrow \mathcal{Z}_0 + E_{sentence}, \tag{12}$$

where $E_{sentence} \in \mathbb{R}^{(n+1) \times d}$ are the sentence position encodings. As for the visual words in a sentence, a word position encoding is added to each word embedding:

$$Y_0^i \leftarrow Y_0^i + E_{word}, \ i = 1, 2, \cdots, n \tag{13}$$

where $E_{word} \in \mathbb{R}^{m \times c}$ are the word position encodings which are shared across sentences. In this way, sentence position encoding can maintain the global spatial information, while word position encoding is used for preserving the local relative position.

## 2.3 Complexity Analysis

A standard transformer block includes two parts, *i.e.*, the multi-head self-attention and multi-layer perceptron. The FLOPs of MSA are $2nd(d_k + d_v) + n^2(d_k + d_v)$, and the FLOPs of MLP are $2nd_v r d_v$ where $r$ is the dimension expansion ratio of hidden layer in MLP. Overall, the FLOPs of a standard transformer block are

$$\text{FLOPs}_T = 2nd(d_k + d_v) + n^2(d_k + d_v) + 2nddr. \tag{14}$$

Since $r$ is usually set as 4, and the dimensions of input, key (query) and value are usually set as the same, the FLOPs calculation can be simplified as

$$\text{FLOPs}_T = 2nd(6d + n). \tag{15}$$

The number of parameters can be obtained as

$$\text{Params}_T = 12dd. \tag{16}$$

Our TNT block consists of three parts: an inner transformer block $T_{in}$, an outer transformer block $T_{out}$ and a linear layer. The computation complexity of $T_{in}$ and $T_{out}$ are $2nmc(6c + m)$ and $2nd(6d + n)$ respectively. The linear layer has FLOPs of $nmcd$. In total, the FLOPs of TNT block are

$$\text{FLOPs}_{TNT} = 2nmc(6c + m) + nmcd + 2nd(6d + n). \tag{17}$$

Similarly, the parameter complexity of TNT block is calculated as

$$\text{Params}_{TNT} = 12cc + mcd + 12dd. \tag{18}$$

Although we add two more components in our TNT block, the increase of FLOPs is small since $c \ll d$ and $\mathcal{O}(m) \approx \mathcal{O}(n)$ in practice. For example, in the DeiT-S configuration, we have $d = 384$ and $n = 196$. We set $c = 24$ and $m = 16$ in our structure of TNT-S correspondingly. From Eq. 15 and Eq. 17, we can obtain that $\text{FLOPs}_T = 376M$ and $\text{FLOPs}_{TNT} = 429M$. The FLOPs ratio of TNT block over standard transformer block is about $1.14\times$. Similarly, the parameters ratio is about $1.08\times$. With a small increase of computation and memory cost, our TNT block can efficiently model the local structure information and achieve a much better trade-off between accuracy and complexity as demonstrated in the experiments.

## 2.4 Network Architecture

We build our TNT architectures by following the basic configuration of ViT [9] and DeiT [31]. The patch size is set as 16×16. The number of sub-patches is set as $m = 4 \cdot 4 = 16$ by default. Other size values are evaluated in the ablation studies. As shown in Table 1, there are three variants of TNT networks with different model sizes, namely, TNT-Ti, TNT-S and TNT-B. They consist of 6.1M, 23.8M and 65.6M parameters respectively. The corresponding FLOPs for processing a 224×224 image are 1.4B, 5.2B and 14.1B respectively.

Table 1: Variants of our TNT architecture. 'Ti' means tiny, 'S' means small, and 'B' means base. The FLOPs are calculated for images at resolution 224×224.

| Model | Depth | Inner transformer | | | Outer transformer | | | Params | FLOPs |
| | | dim $c$ | #heads | MLP $r$ | dim $d$ | #heads | MLP $r$ | (M) | (B) |
|---|---|---|---|---|---|---|---|---|---|
| TNT-Ti | 12 | 12 | 2 | 4 | 192 | 3 | 4 | 6.1 | 1.4 |
| TNT-S | 12 | 24 | 4 | 4 | 384 | 6 | 4 | 23.8 | 5.2 |
| TNT-B | 12 | 40 | 4 | 4 | 640 | 10 | 4 | 65.6 | 14.1 |

# 3 Experiments

In this section, we conduct extensive experiments on visual benchmarks to evaluate the effectiveness of the proposed TNT architecture.

Table 2: Details of used visual datasets.

| Dataset | Type | Train size | Val size | #Classes |
|---|---|---|---|---|
| ImageNet [26] | Pretrain | 1,281,167 | 50,000 | 1,000 |
| Oxford 102 Flowers [22] | | 2,040 | 6,149 | 102 |
| Oxford-IIIT Pets [23] | | 3,680 | 3,669 | 37 |
| iNaturalist 2019 [34] | Classification | 265,240 | 3,003 | 1,010 |
| CIFAR-10 [16] | | 50,000 | 10,000 | 10 |
| CIFAR-100 [16] | | 50,000 | 10,000 | 100 |
| COCO2017 [19] | Detection | 118,287 | 5,000 | 80 |
| ADE20K [45] | Segmentation | 20,210 | 2,000 | 150 |

### 3.1 Datasets and Experimental Settings

**Datasets.** ImageNet ILSVRC 2012 [26] is an image classification benchmark consisting of 1.2M training images belonging to 1000 classes, and 50K validation images with 50 images per class. We adopt the same data augmentation strategy as that in DeiT [31] including random crop, random clip, Rand-Augment [7], Random Erasing [44], Mixup [42] and CutMix [41]. For the license of ImageNet dataset, please refer to http://www.image-net.org/download.

In addition to ImageNet, we also test on the downstream tasks with transfer learning to evaluate the generalization ability of TNT. The details of used visual datasets are listed in Table 2. The data augmentation strategy of image classification datasets are the same as that of ImageNet. For COCO and ADE20K, the data augmentation strategy follows that in PVT [36]. For the licenses of these datasets, please refer to the original papers.

**Implementation Details.** We utilize the training strategy provided in DeiT [31]. The main advanced technologies apart from common settings [12] include AdamW [20], label smoothing [27], DropPath [18], and repeated augmentation [14]. We list the hyper-parameters in Table 3 for better understanding. All the models are implemented with PyTorch [24] and MindSpore [15] and trained on NVIDIA Tesla V100 GPUs. The potential negative societal impacts may include energy consumption and carbon dioxide emissions of GPU computation.

Table 3: Default training hyper-parameters used in our method, unless stated otherwise.

| Epochs | Optimizer | Batch size | Learning rate | LR decay | Weight decay | Warmup epochs | Label smooth | Drop path | Repeated Aug |
|---|---|---|---|---|---|---|---|---|---|
| 300 | AdamW | 1024 | 1e-3 | cosine | 0.05 | 5 | 0.1 | 0.1 | $\sqrt{}$ |

### 3.2 TNT on ImageNet

We train our TNT models with the same training settings as that of DeiT [31]. The recent transformer-based models like ViT [9] and DeiT [31] are compared. To have a better understanding of current progress of visual transformers, we also include the representative CNN-based models such as ResNet [12], RegNet [25] and EfficientNet [28]. The results are shown in Table 4. We can see that our transformer-based model, *i.e.*, TNT outperforms all other visual transformer models. In particular, TNT-S achieves 81.5% top-1 accuracy which is 1.7% higher than the baseline model DeiT-S, indicating the benefit of the introduced TNT framework to preserve local structure information inside the patch. Compared to CNNs, TNT can outperform the widely-used ResNet and RegNet. Note that all the transformer-based models are still inferior to EfficientNet which utilizes special depth-wise convolutions, so it is yet a challenge of how to beat EfficientNet using pure transformer.

Table 4: Results of TNT and other networks on ImageNet.

| Model | Resolution | Params (M) | FLOPs (B) | Top-1 | Top-5 |
|---|---|---|---|---|---|
| **CNN-based** | | | | | |
| ResNet-50 [12] | 224×224 | 25.6 | 4.1 | 76.2 | 92.9 |
| ResNet-152 [12] | 224×224 | 60.2 | 11.5 | 78.3 | 94.1 |
| RegNetY-8GF [25] | 224×224 | 39.2 | 8.0 | 79.9 | - |
| RegNetY-16GF [25] | 224×224 | 83.6 | 15.9 | 80.4 | - |
| EfficientNet-B3 [28] | 300×300 | 12.0 | 1.8 | 81.6 | 94.9 |
| EfficientNet-B4 [28] | 380×380 | 19.0 | 4.2 | 82.9 | 96.4 |
| **Transformer-based** | | | | | |
| DeiT-Ti [31] | 224×224 | 5.7 | 1.3 | 72.2 | - |
| TNT-Ti | 224×224 | 6.1 | 1.4 | **73.9** | **91.9** |
| DeiT-S [31] | 224×224 | 22.1 | 4.6 | 79.8 | - |
| PVT-Small [36] | 224×224 | 24.5 | 3.8 | 79.8 | - |
| T2T-ViT_t-14 [40] | 224×224 | 21.5 | 5.2 | 80.7 | - |
| TNT-S | 224×224 | 23.8 | 5.2 | **81.5** | **95.7** |
| ViT-B/16 [9] | 384×384 | 86.4 | 55.5 | 77.9 | - |
| DeiT-B [31] | 224×224 | 86.4 | 17.6 | 81.8 | - |
| T2T-ViT_t-24 [40] | 224×224 | 63.9 | 13.2 | 82.2 | - |
| TNT-B | 224×224 | 65.6 | 14.1 | **82.9** | **96.3** |

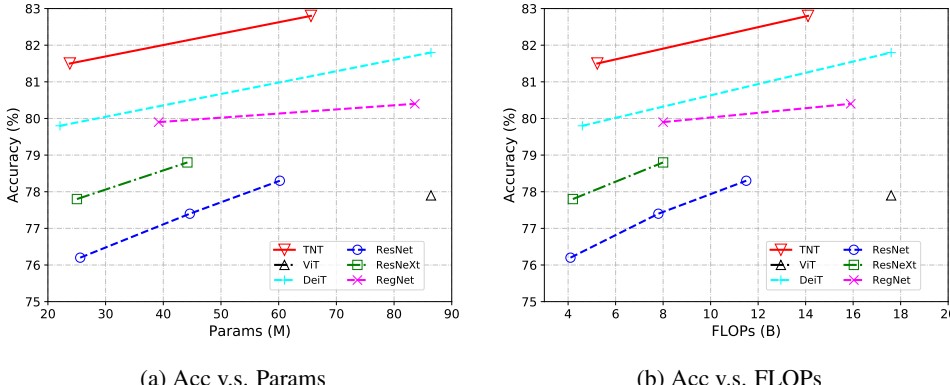

|                      |                      |
|:--------------------:|:--------------------:|
| (a) Acc v.s. Params  | (b) Acc v.s. FLOPs   |

Figure 2: Performance comparison of the representative visual backbone networks on ImageNet.

We also plot the accuracy-parameters and accuracy-FLOPs line charts in Fig. 2 to have an intuitive comparison of these models. Our TNT models consistently outperform other transformer-based models by a significant margin.

**Inference speed.**  Deployment of transformer models on devices is important for practical applications, so we test the inference speed of our TNT model. Following [31], the throughput is measured on an NVIDIA V100 GPU and PyTorch, with 224×224 input size. Since the resolution and content inside the patch is smaller than that of the whole image, we may need fewer blocks to learn its representation. Thus, we can reduce the used TNT blocks and replace some with vanilla transformer blocks. From the results in Table 5, we can see that our TNT is more efficient than DeiT and PVT by achieving higher accuracy with similar inference speed.

Table 5: GPU throughput comparison of vision transformer models.

| Model | Indices of TNT blocks | FLOPs (B) | Throughput (images/s) | Top-1 |
|-------|:---------------------:|:---------:|:---------------------:|:-----:|
| DeiT-S [31]    | -                                 | 4.6  | 907 | 79.8 |
| DeiT-B [31]    | -                                 | 17.6 | 292 | 81.8 |
| PVT-Small [36] | -                                 | 3.8  | 820 | 79.8 |
| PVT-Medium [36]| -                                 | 6.7  | 526 | 81.2 |
| TNT-S   | [1, 2, 3, 4, 5, 6, 7, 8, 9, 10, 11, 12] | 5.2  | 428 | 81.5 |
| TNT-S-1 | [1, 4, 8, 12]                     | 4.8  | 668 | 81.4 |
| TNT-S-2 | [1, 6, 12]                        | 4.7  | 704 | 81.3 |
| TNT-S-3 | [1, 6]                            | 4.7  | 757 | 81.1 |
| TNT-S-4 | [1]                               | 4.6  | 822 | 80.8 |

## 3.3  Ablation Studies

**Effect of position encodings.**  Position information is important for image recognition. In TNT structure, sentence position encoding is for maintaining global spatial information, and word position encoding is used to preserve locally relative position. We verify their effect by removing them separately. As shown in Table 6, we can see that TNT-S with both patch position encoding and word position encoding performs the best by achieving 81.5% top-1 accuracy. Removing sentence/word position encoding results in a 0.8%/0.7% accuracy drop respectively, and removing all position encodings heavily decrease the accuracy by 1.0%.

Table 6: Effect of position encoding.

| Model | Position encoding | | Top-1 |
|-------|:---------------:|:---------:|:-----:|
|       | Sentence-level  | Word-level |       |
| TNT-S | ✗ | ✗ | 80.5 |
|       | ✔ | ✗ | 80.8 |
|       | ✗ | ✔ | 80.7 |
|       | ✔ | ✔ | 81.5 |

**Number of heads.**  The effect of #heads in standard transformer has been investigated in multiple works [21, 35] and a head width of 64 is recommended for visual tasks [9, 31]. We adopt the head width of 64 in outer transformer block in our model. The number of heads in inner transformer block

is another hyper-parameter for investigation. We evaluate the effect of #heads in inner transformer block (Table 7). We can see that a proper number of heads (*e.g.*, 2 or 4) achieve the best performance.

Table 7: Effect of #heads in inner transformer block in TNT-S.

| #heads | 1 | 2 | 4 | 6 | 8 |
|---|---|---|---|---|---|
| Top-1 | 81.0 | 81.4 | 81.5 | 81.3 | 81.1 |

**Number of visual words.** In TNT, the input image is split into a number of 16×16 patches and each patch is further split into $m$ sub-patches (visual words) of size $(s, s)$ for computational efficiency. Here we test the effect of hyper-parameter $m$ on TNT-S architecture. When we change $m$, the embedding dimension $c$ also changes correspondingly to control the FLOPs. As shown in Table 8, we can see that the value of $m$ has slight influence on the performance, and we use $m = 16$ by default for its efficiency, unless stated otherwise.

Table 8: Effect of #words $m$.

| $m$ | $c$ | Params | FLOPs | Top-1 |
|---|---|---|---|---|
| 64 | 6 | 23.8M | 5.1B | 81.0 |
| 16 | 24 | 23.8M | 5.2B | 81.5 |
| 4 | 96 | 25.1M | 6.0B | 81.1 |

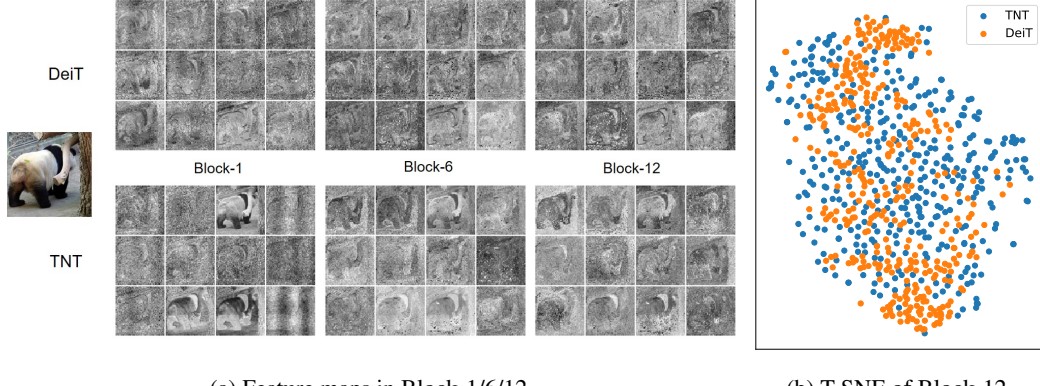

(a) Feature maps in Block-1/6/12.  (b) T-SNE of Block-12.

Figure 3: Visualization of the features of DeiT-S and TNT-S.

## 3.4 Visualization

**Visualization of Feature Maps.** We visualize the learned features of DeiT and TNT to further understand the effect of the proposed method. For better visualization, the input image is resized to 1024×1024. The feature maps are formed by reshaping the patch embeddings according to their spatial positions. The feature maps in the 1-st, 6-th and 12-th blocks are shown in Fig. 3(a) where 12 feature maps are randomly sampled for these blocks each. In TNT, the local information are better preserved compared to DeiT. We also visualize all the 384 feature maps in the 12-th block using t-SNE [33] (Fig. 3(b)). We can see that the features of TNT are more diverse and contain richer information than those of DeiT. These benefits owe to the introduction of inner transformer block for modeling local features.

In addition to the patch-level features, we also visualize the pixel-level embeddings of TNT in Fig. 4. For each patch, we reshape the word embeddings according to their spatial positions to form the feature maps and then average these feature maps by the channel dimension. The averaged feature maps corresponding to the 14×14 patches are shown in Fig. 4. We can see that the local information is well preserved in the shallow layers, and the representations become more abstract gradually as the network goes deeper.

**Visualization of Attention Maps.** There are two self-attention layers in our TNT block, *i.e.*, an inner self-attention and an outer self-attention for modeling relationship among visual words and sentences respectively. We show the attention maps of different queries in the inner transformer in Figure 5. For a given query visual word, the attention values of visual words with similar appearance are higher, indicating their features will be interacted more relevantly with the query. These interactions are missed in ViT and DeiT, *etc*. The attention maps in the outer transformer can be found in the supplemental material.

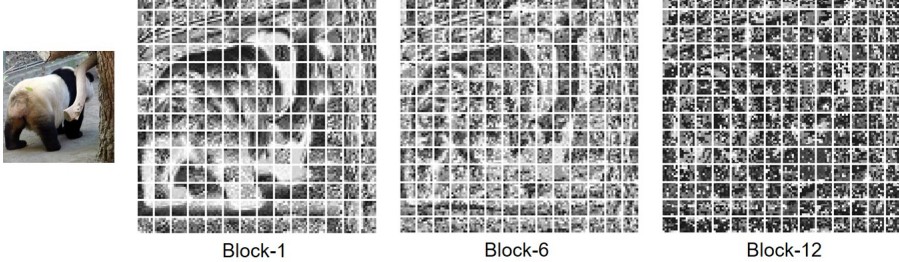

|  |  |  |
|:-:|:-:|:-:|
| Block-1 | Block-6 | Block-12 |

Figure 4: Visualization of the averaged word embeddings of TNT-S.



Figure 5: Attention maps of different queries in the inner transformer. Red cross symbol denotes the query location.

## 3.5 Transfer Learning

To demonstrate the strong generalization ability of TNT, we transfer TNT-S, TNT-B models trained on ImageNet to the downstream tasks.

**Pure Transformer Image Classification.** Following DeiT [31], we evaluate our models on 4 image classification datasets with training set size ranging from 2,040 to 50,000 images. These datasets include superordinate-level object classification (CIFAR-10 [16], CIFAR-100 [16]) and fine-grained object classification (Oxford-IIIT Pets [23], Oxford 102 Flowers [22] and iNaturalist 2019 [34]), shown in Table 2. All models are fine-tuned with an image resolution of 384×384. We adopt the same training settings as those at the pre-training stage by preserving all data augmentation strategies. In order to fine-tune in a different resolution, we also interpolate the position embeddings of new patches. For CIFAR-10 and CIFAR-100, we fine-tune the models for 64 epochs, and for fine-grained datasets, we fine-tune the models for 300 epochs. Table 9 compares the transfer learning results of TNT to those of ViT, DeiT and other convolutional networks. We find that TNT outperforms DeiT in most datasets with less parameters, which shows the superiority of modeling pixel-level relations to get better feature representation.

Table 9: Results on downstream image classification tasks with ImageNet pre-training. $\uparrow 384$ denotes fine-tuning with 384×384 resolution.

| Model | Params (M) | ImageNet | CIFAR10 | CIFAR100 | Flowers | Pets | iNat-19 |
|---|---|---|---|---|---|---|---|
| **CNN-based** | | | | | | | |
| Grafit ResNet-50 [32] | 25.6 | 79.6 | - | - | 98.2 | - | 75.9 |
| Grafit RegNetY-8GF [32] | 39.2 | - | - | - | 99.1 | - | 80.0 |
| EfficientNet-B5 [28] | 30 | 83.6 | 98.7 | 91.1 | 98.5 | - | - |
| **Transformer-based** | | | | | | | |
| ViT-B/16$_{\uparrow 384}$ [9] | 86.4 | 77.9 | 98.1 | 87.1 | 89.5 | 93.8 | - |
| DeiT-B$_{\uparrow 384}$ [31] | 86.4 | 83.1 | 99.1 | 90.8 | 98.4 | - | - |
| TNT-S$_{\uparrow 384}$ | 23.8 | 83.1 | 98.7 | 90.1 | 98.8 | 94.7 | 81.4 |
| TNT-B$_{\uparrow 384}$ | 65.6 | **83.9** | **99.1** | **91.1** | **99.0** | **95.0** | **83.2** |

**Pure Transformer Object Detection.** We construct a pure transformer object detection pipeline by combining our TNT and DETR [3]. For fair comparison, we adopt the training and testing settings in PVT [36] and add a 2×2 average pooling to make the output size of TNT backbone the same as that of PVT and ResNet. All the compared models are trained using AdamW [20] with batch size of 16 for 50 epochs. The training images are randomly resized to have a shorter side in the range of [640,800] and a longer side within 1333 pixels. For testing, the shorter side is set as 800 pixels. The results on COCO val2017 are shown in Table 10. Under the same setting, DETR with TNT-S backbone outperforms the representative pure transformer detector DETR+PVT-Small by 3.5 AP with similar parameters.

Table 10: Results of object detection on COCO2017 val set with ImageNet pre-training. [†]Results from our implementation.

| Backbone | Params | Epochs | AP | $AP_{50}$ | $AP_{75}$ | $AP_S$ | $AP_M$ | $AP_L$ |
|---|---|---|---|---|---|---|---|---|
| ResNet-50 [36] | 41M | 50 | 32.3 | 53.9 | 32.3 | 10.7 | 33.8 | 53.0 |
| DeiT-S[†] [31] | 38M | 50 | 33.9 | 54.7 | 34.3 | 11.0 | 35.4 | 56.6 |
| PVT-Small [36] | 40M | 50 | 34.7 | 55.7 | 35.4 | 12.0 | 36.4 | 56.7 |
| PVT-Medium [36] | 57M | 50 | 36.4 | 57.9 | 37.2 | 13.0 | 38.7 | 59.1 |
| TNT-S | 39M | 50 | **38.2** | **58.9** | **39.4** | **15.5** | **41.1** | **58.8** |

**Pure Transformer Semantic Segmentation.** We adopt the segmentation framework of Trans2Seg [38] to build the pure transformer semantic segmentation based on TNT backbone. We follow the training and testing configuration in PVT [36] for fair comparison. All the compared models are trained by AdamW optimizer with initial learning rate of 1e-4 and polynomial decay schedule. We apply random resize and crop of $512 \times 512$ during training. The ADE20K results with single scale testing are shown in Table 11. With similar parameters, Trans2Seg with TNT-S backbone achieves 43.6% mIoU, which is 1.0% higher than that of PVT-small backbone and 2.8% higher than that of DeiT-S backbone.

Table 11: Results of semantic segmentation on ADE20K val set with ImageNet pre-training. [†]Results from our implementation.

| Backbone | Params | FLOPs | Steps | mIoU |
|---|---|---|---|---|
| ResNet-50 [38] | 56.1M | 79.3G | 40k | 39.7 |
| DeiT-S[†] [31] | 30.3M | 27.2G | 40k | 40.5 |
| PVT-Small [43] | 32.1M | 31.6G | 40k | 42.6 |
| TNT-S | 32.1M | 30.4G | 40k | **43.6** |

## 4   Conclusion

In this paper, we propose a novel Transformer-iN-Transformer (TNT) network architecture for visual recognition. In particular, we uniformly split the image into a sequence of patches (visual sentences) and view each patch as a sequence of sub-patches (visual words). We introduce a TNT block in which an outer transformer block is utilized for processing the sentence embeddings and an inner transformer block is used to model the relation among word embeddings. The information of visual word embeddings is added to the visual sentence embedding after the projection of a linear layer. We build our TNT architecture by stacking the TNT blocks. Compared to the conventional vision transformers (ViT) which corrupts the local structure of the patch, our TNT can better preserve and model the local information for visual recognition. Extensive experiments on ImageNet and downstream tasks have demonstrate the effectiveness of the proposed TNT architecture.

## Acknowledgement

This work was supported by NSFC (62072449, 61632003), Guangdong-Hongkong-Macao Joint Research Grant (2020B1515130004) and Macao FDCT (0018/2019/AKP, 0015/2019/AKP).

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
