# Transformer in Transformer
# Supplemental Material

**Kai Han**[1,2] **An Xiao**[2] **Enhua Wu**[1,3*] **Jianyuan Guo**[2] **Chunjing Xu**[2] **Yunhe Wang**[2*]
[1]State Key Lab of Computer Science, ISCAS & UCAS
[2]Huawei Noah's Ark Lab
[3]University of Macau
{hankai,weh}@ios.ac.cn, yunhe.wang@huawei.com

## 1   Visualization of Attention Maps

**Attention between Patches.**   In Figure 1, we plot the attention maps from each patch to all the patches. We can see that for both DeiT-S and TNT-S, more patches are related as layer goes deeper. This is because the information between patches has been fully communicated with each other in deeper layers. As for the difference between DeiT and TNT, the attention of TNT can focus on the meaningful patches in Block-12, while DeiT still pays attention to the tree which is not related to the pandas.

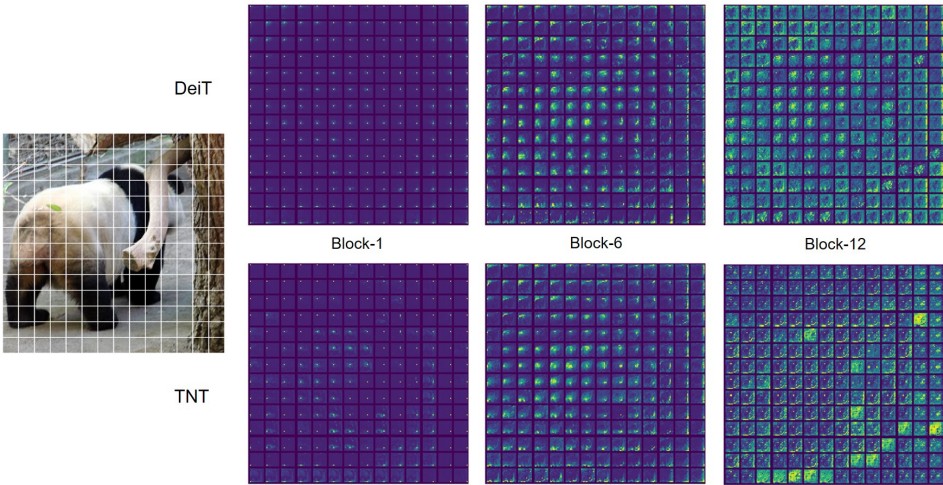

Figure 1: Visualization of the attention maps between all patches in outer transformer block.

**Attention between Class Token and Patches.**   In Figure 2, we plot the attention maps between class token to all the patches for some randomly sampled images. We can see that the output feature mainly focus on the patches related to the object to be recognized.

## 2   Exploring SE module in TNT

Inspired by squeeze-and-excitation (SE) network for CNNs [2], we propose to explore channel-wise attention for transformers. We first average all the sentence (word) embeddings and use a two-layer

---

*Corresponding author.

35th Conference on Neural Information Processing Systems (NeurIPS 2021).

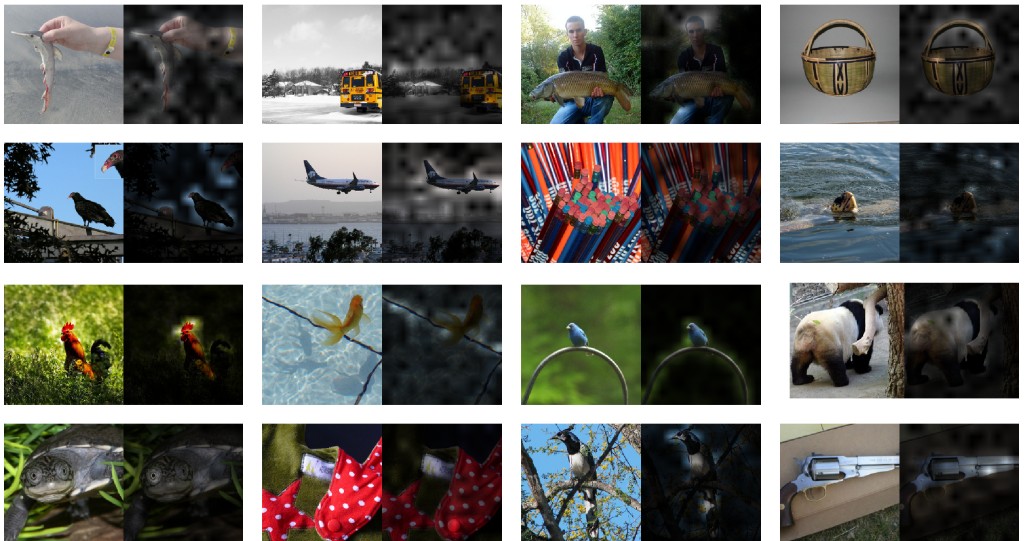

Figure 2: Example attention maps from the output token to the input space.

MLP to calculate the attention values. The attention is multiplied to all the embeddings. The SE module only brings in a few extra parameters but is able to perform dimension-wise attention for feature enhancement. From the results in Table 1, adding SE module into TNT can further improve the accuracy slightly.

Table 1: Exploring SE module in TNT.

| Model | Resolution | Params (M) | FLOPs (B) | Top-1 (%) | Top-5 (%) |
|---|---|---|---|---|---|
| TNT-S | 224×224 | 23.8 | 5.2 | 81.5 | 95.7 |
| TNT-S + SE | 224×224 | 24.7 | 5.2 | 81.7 | 95.7 |

## 3    Object Detection with Faster RCNN

As a general backbone network, TNT can also be applied with multi-scale vision models like Faster RCNN [4]. We extract the features from different layers of TNT to construct multi-scale features. In particular, FPN takes 4 levels of features ($\frac{1}{4}$, $\frac{1}{8}$, $\frac{1}{16}$, $\frac{1}{32}$) as input, while the resolution of feature of every TNT block is $\frac{1}{16}$. We select the 4 layers from shallow to deep (3rd, 6th, 9th, 12th) to form multi-level representation. To match the feature shape, we insert deconvolution/convolution layers with proper stride. We evaluate TNT-S and DeiT-S on Faster RCNN with FPN [3]. The DeiT model is used in the same way. The COCO2017 val results are shown in Table 2. TNT achieves much better performance than ResNet and DeiT backbones, indicating its generalization for FPN-like framework.

Table 2: Results of Faster RCNN object detection on COCO minival set with ImageNet pre-training. [†]Results from our implementation.

| Backbone | Params (M) | Epochs | AP | $AP_{50}$ | $AP_{75}$ | $AP_S$ | $AP_M$ | $AP_L$ |
|---|---|---|---|---|---|---|---|---|
| ResNet-50 [3, 1] | 41.5 | 12 | 37.4 | 58.1 | 40.4 | 21.2 | 41.0 | 48.1 |
| DeiT-S[†] [5] | 46.4 | 12 | 39.9 | 62.8 | 42.6 | 23.4 | 42.5 | 54.0 |
| TNT-S | 48.1 | 12 | 41.5 | 64.1 | 44.5 | 25.7 | 44.6 | 55.4 |