# OpenReview forum: "Transformer in Transformer"
_NeurIPS.cc/2021/Conference — NeurIPS 2021 Poster_

### Official Review · Reviewer_u1uh · 2021-07-03

**Rating:** 7
**Confidence:** 4

**Summary:**

This paper proposes an improved ViT network that applies self-attention layers to a hierarchical patches: a 16x16 patch on the original image (the authors called them "sentence-level attention), and a 4x4 sub patch for each of the original path (the authors called them "word-level attention). Overall, the idea is very simple and the results are promising: 1.7% better top-1 accuracy on ImageNet than DeiT.

**Ethics Review Area:**

["I don’t know"]

**Main Review:**

----------- Strengthes -----------------

(1) Very simple but effective idea: the 16x16 patch on the original ViT paper seems like an obvious problem, as it brutally combines a large image patch into a single "word" in order to feed to Transformer.  Therefore, this paper proposes to apply intra-word attention to solve this issue. It is a quite nice idea to me.
(2) Comprehensive analysis: it provides detailed analysis on the additional FLOPs/latency, and provides various analysis on visualization and transfer learning.
(3) Promising results: compared to DeiT, the improvements are promising.


--------- Weaknesses ------------------

(1) Missing latency results: while the FLOPs/Parameters overhead seem to be small, it is unclear whether such intra-work attention is slow or not on real hardwares like GPUs/TPUs. It would be helpful to compare the training/inference speed.
(2) Fair comparison: ViT is generally sensitive to training hyperparamters (e.g., DeiT has much better accuracy than ViT for the same network). Although the authors report 1.7% better accuracy, but it is not clear whether such improvements are partially from training settings. I know the authors use similar settings as DeiT, but it is better to report your reproduced numbers of ViT/DeiT using EXACTLY the same settings as your TNT.


-------  Additional questions ----------

When the input image is small, using two-levels of attention is probably good enough. However, if the input image is large (like 640 for detection or 1024 for segmentation), would it be reasonable to use more levels of attention?

**Time Spent Reviewing:**

4

---

> ### Author Response · Authors · 2021-08-09
> **Authors' response**
>
> We thank the reviewer for detailed comments.
>
> **Q1**: Missing latency results: ... It would be helpful to compare the training/inference speed.
>
> **A2**: Thanks for the nice suggestion.  We'll discuss the CPU/GPU latency of the proposed TNT architecture in the updated version.
>
> First, the inference speed on CPU is satisfied as shown in the table. The latency is measured on an Intel CPU with batch size of 1.
> Compared with the baseline DeiT, the proposed TNT can achieve a better latency/acc trade-off.
>
> | Model  | FLOPs (B) | CPU latency (s) | ImageNet Top-1 |
> | ------ | --------- | --------------- | ----- |
> | TNT-S  | 5.2       | 5.9             | 81.5  |
> | TNT-B  | 14.1      | 9.7             | 82.8  |
> | DeiT-S | 4.6       | 4.7             | 79.8  |
> | DeiT-B | 17.6      | 9.9             | 81.8  |
>
> Second, since we introduce some additional subtle operations and calculations, which cannot well utilize the computing power of GPU. Thus, the GPU latency of TNT is worse than that of DeiT. However, the training/inference speed on GPU can be further optimized in the engineering level or model level.
>
> 1. The current implementation is based on PyTorch which is not optimal for speed. To achieve higher speed on GPU/TPU, more engineering works like CUDA implementation or hardware unit support, are required to handle imbalanced matrix multiplication. For the common settings of DeiT-S/TNT-S, $n=196$, $d=384$, $m=16$ and $c=24$. In the outer transformer, the typical matrix multiplication is $\mathbb{R}^{196\times 384} \times \mathbb{R}^{384\times 384}$, with FLOPs of 28,901,376. While in the inner transformer, the typical matrix multiplication is $\mathbb{R}^{(196*16)\times 24} \times \mathbb{R}^{24\times 24}$, with FLOPs of 1,806,336. So the shape of matrix in the inner transformer is extremely imbalanced and we will continue to optimize in our future work.
> 2. We can optimize the speed in the model level. Since the resolution and content inside the patch is smaller than that of the whole image, we may need fewer blocks to learn its representation. Thus, we can reduce the used TNT blocks and replace some with vanilla transformer blocks. As shown in the table, we can achieve a better accuracy-speed trade-off by using fewer TNT blocks. For example, TNT-S-1 with 4 TNT blocks can achieve 81.4 ImageNet Top-1 with the inference speed of 668 images/s. The accuracy-speed trade-off is shown in this figure (https://i.postimg.cc/RhCbKZ88/GPU-latency.png).
>
> | Model           | Indices of TNT blocks        | FLOPs (B) | Training speed (images/s, 8\*V100) | Inference speed (images/s, V100) | ImageNet Top-1 |
> | --------------- | ---------------------------- | --------- | --------------------------------- | -------------------------------- | ----- |
> | TNT-S           | [1,2,3,4,5,6,7,8,9,10,11,12] | 5.2       | ~1000                              | 428                              | 81.5  |
> | TNT-S-1         | [1,4,8,12]                   | 4.8       | ~1900                              | 668                              | 81.4  |
> | TNT-S-2         | [1,6,12]                     | 4.7       | ~2100                              | 704                              | 81.3  |
> | TNT-S-3         | [1,6]                        | 4.7       | ~2400                              | 757                              | 81.1  |
> | TNT-S-4         | [1]                          | 4.6       | ~2900                              | 822                              | 80.8  |
> | TNT-S-5         | [6]                          | 4.6       | ~2900                              | 822                              | 80.8  |
> | TNT-S-6         | [12]                         | 4.6       | ~2900                              | 822                              | 80.6  |
> | DeiT-S          | -                            | 4.6       | ~3200                              | 907                              | 79.8  |
> | PVT-Small [r1]  | -                            | 3.8       | -                                 | 820                              | 79.8  |
> | PVT-Medium [r1] | -                            | 6.7       | -                                 | 526                              | 81.2  |
> | DeiT-B          | -                            | 17.6      | ~1700                              | 292                              | 81.8  |
>
> [r1] Wang, Wenhai, et al. "Pyramid vision transformer: A versatile backbone for dense prediction without convolutions." ICCV 2021.
>
>
> **Q2**: Fair comparison: ViT is generally sensitive to ... the same settings as your TNT.
>
> **A2**: We have trained and tested TNT models using the same settings as the DeiT paper by our implementation. The reproduced result of DeiT is very close to that in the DeiT paper as shown in the following table:
>
>   | Model  | Training script                                              | ImageNet Top-1 |
>   | ------ | ------------------------------------------------------------ | ----- |
>   | DeiT-S | DeiT paper (AdamW, batch size=1024, epochs=300, lr=0.001, weight decay=0.05, label smoothing, stoch. depth=0.1, rand augment, Mixup, CutMix) | 79.8  |
>   | DeiT-S | Our implementation (AdamW, batch size=1024, epochs=300, lr=0.001, weight decay=0.05, label smoothing, stoch. depth=0.1, rand augment, Mixup, CutMix) | 79.9  |
>   | TNT-S  | Our implementation (AdamW, batch size=1024, epochs=300, lr=0.001, weight decay=0.05, label smoothing, stoch. depth=0.1, rand augment, Mixup, CutMix) | 81.5  |
>
>
> **Q3**: When the input image is small, ... would it be reasonable to use more levels of attention?
>
> **A3**: We have evaluated TNT backbone on detection and segmentation tasks. The two-levels of attention can enhance the performance by a significant margin. Using more levels of attention for larger input image is reasonable and is a good direction for future investigation.

---

> > ### Comment · Reviewer_u1uh · 2021-08-18
> > **Thanks for the response!**
> >
> > Thanks for the response for mostly addressing my concerns.
> >
> > I agree that latency can be optimized in engineering and model level, but people may not adopt the new models if models are slow on real hardware. I appreciate the additional results, which show TNT's speed is indeed promising.
> >
> > Additional question: should the x-axis in your figure https://i.postimg.cc/RhCbKZ88/GPU-latency.png  be "speed" instead of "latency"?
> >
> > I recommend accepting this paper.

---

> > > ### Author Response · Authors · 2021-08-18
> > > **Authors' response**
> > >
> > > Thanks for the addtional question. The x-axis in our figure is speed (images/s) instead of latency (s). The fixed figure is availble at https://i.postimg.cc/XNwr7TqD/GPU-speed.png.

---

### Official Review · Reviewer_6Dax · 2021-07-14

**Rating:** 7
**Confidence:** 4

**Summary:**

This paper proposes a novel vision transformer (TNT) to better model the fine-grained relationship within local image patches.
Specifically, based on the patch splitting scheme in ViT, the authors further split the local patches ("visual sentence") into smaller ones ("visual words"). Inner transformers are adopted to model relationships between words and improve sentence representation.
Experiments on ImageNet classification and several downstream tasks validate the effectiveness of the proposed method.

**Limitations And Societal Impact:**

The authors have mentioned potential negative impacts such as energy consumption and carbon dioxide emissions of GPU computation. The limitations are also mentioned in the Experiments section.

**Main Review:**

Strength:
1. The motivation is clear. The idea of modeling fine-grained local features is also explored in concurrent works, e.g., Swin Transformer[1] and Twins[2].
2. Experiments are carried out for ImageNet classification and several downstream tasks.
3. The writing is good and easy to follow.


Weakness:
1. With the inner transformers, the number of transformer layers in TNT is doubled compared to the DeiT baseline. It's hard to judge whether the performance improvement owes to better intra-visual sentence relationship modeling, or simply improved model depth. An empirical study is recommended to figure out the reason.

2. Compared with concurrent works, the proposed method only brings moderate performance gain to the DeiT baseline. Moreover, only two variants of TNT are instantiated, making it hard to understand the scaling property of TNT or compare with other methods in a systematic way. Considering the difficulty of training deep ViTs [3, 4] and the doubled model depth in TNT, will the model be more prone to overfitting when the model scales? Besides, the performance of lightweight backbone architecture is also of great importance, especially for applications with limited computation resources. Therefore, smaller and larger models, e.g., ~6M (Tiny), ~100M (Large), or bigger ones, should be trained and evaluated for a comprehensive comparison.

3. Modern object detectors [5, 6] usually exploit multi-scale features, where small objects are detected from high-resolution feature maps. However, there is no design in TNT to make it adaptable to high-resolution inputs or features. This will inevitably result in high memory consumption. In the experiments, TNT-based object detection and segmentation can only be performed on single-scale image features. The reviewer wonders that is it adaptable to methods that can significantly benefit from multi-scale features, e.g., Deformable DETR and Faster RCNN with FPN, considering that the TNT is designed as a general backbone network for different downstream tasks?

4. Since TNT specifically focuses on modeling the "visual words" by splitting the local patches ("visual sentence") into smaller ones, it should provide better local feature representations, which could be useful for fine-grained visual understanding tasks, e.g., fine-grained image classification. Although the authors have carried experiments on the Flowers and Pets benchmarks, the performance on these two benchmarks has been saturated. Experiments on more challenging datasets such as iNaturalist19 can be provided to further evaluate the effectiveness of the proposed method.

References:
1. Swin Transformer: Hierarchical Vision Transformer using Shifted Windows
2. Twins: Revisiting the Design of Spatial Attention in Vision Transformers
3. An Image is Worth 16x16 Words: Transformers for Image Recognition at Scale
4. Going deeper with Image Transformers
5. Deformable DETR: Deformable Transformers for End-to-End Object Detection
6. Feature Pyramid Networks for Object Detection

**Time Spent Reviewing:**

10

---

> ### Author Response · Authors · 2021-08-10
> **Authors' response**
>
>
> We thank the reviewer for detailed comments.
>
> **Q1**: With the inner transformers, ... An empirical study is recommended to figure out the reason.
>
> **A1**: Thanks for the nice suggestion. We conduct an empirical study by simply improve model depth of DeiT. For fair comparison, we keep the FLOPs similar and deepen DeiT-S to form DeiT-S-1/2. From the results in the table, we can see that simple deeper models (DeiT-S-1/2) can only affect the accuracy slightly (even worse in DeiT-S-2), which is much lower than that of our TNT-S.
>
> In fact, our TNT is not simply doubling the layers. As shown in Figure 1 in our main paper, the depth of patch-level features is still 12, which is consistent to that of ViT (DeiT). The pixel-level features have an additional 12 layers, which provides extra fine-grained representation for visual recognition. The introduced inner transformer is much smaller than the outer transformer, so it can largely boost the performance with only a few extra parameters.
>
>   | Model    | \#blocks | Embedding dimension | \#params (M) | FLOPs (B) | ImageNet Top-1 |
>   | -------- | -------- | ------------------- | ------------ | --------- | ----- |
>   | DeiT-S   | 12       | 384                 | 22.1         | 4.6       | 79.8  |
>   | DeiT-S-1 | 18       | 336                 | 25.1         | 5.3       | 80.3  |
>   | DeiT-S-2 | 24       | 288                 | 24.5         | 5.3       | 79.6  |
>   | TNT-S    | 12       | 384                 | 23.8         | 5.2       | 81.5  |
>
>
>
> **Q2**: Compared with concurrent works, ... should be trained and evaluated for a comprehensive comparison.
>
> **A2**: Thanks for the nice suggestion. We train the smaller and larger TNT models and show the ImageNet results in the following table. Our TNT models can consistently improve DeiT with a significant margin. For example, TNT-Ti with 5.9M parameters achieves +1.7\% higher accuracy over DeiT-Ti with 5.7M parameters.
>
>   | Model   | \#blocks | Embedding dimension | \#params (M) | FLOPs (B) | ImageNet Top-1    |
>   | ------- | -------- | ------------------- | ------------ | --------- | -------- |
>   | DeiT-Ti | 12       | 192                 | 5.7          | 1.3       | 72.2     |
>   | DeiT-S  | 12       | 384                 | 22.1         | 4.6       | 79.8     |
>   | DeiT-B  | 12       | 768                 | 86.4         | 17.6      | 81.8 |
>   | TNT-Ti  | 12       | 192                 | 5.9          | 1.4       | 73.9     |
>   | TNT-S   | 12       | 384                 | 23.8         | 5.2       | 81.5     |
>   | TNT-B   | 12       | 640                 | 65.6         | 14.1      | 82.8 |
>   | TNT-L-1 | 12       | 768                 | 94.2         | 20.1      | 83.4     |
>   | TNT-L-2 | 16       | 640                 | 97.7         | 21.1      | 83.5     |
>
>
> **Q3**: Modern object detectors [5, 6] ... a general backbone network for different downstream tasks?
>
> **A3**: Thanks for the nice suggestion. We can extract the features from different layers of TNT to construct multi-scale features. In particular, FPN takes 4 levels of features (1/4,1/8,1/16,1/32) as input, while the resolution of feature of every TNT block is 1/16. We select the 4 layers from shallow to deep (3rd,6th,9th,12th) to form multi-level representation. To match the feature shape, we insert deconvolution/convolution layers with proper stride. We evaluate TNT-S and DeiT-S on Faster RCNN with FPN. The DeiT model is used in the same way. The COCO2017 val results are shown in the following table. TNT achieves much better performance than ResNet and DeiT backbones, indicating its generalization for FPN-like framework.
>
>   | Backbone   | \#params (M) | mAP  |
>   | ---------- | ------------ | ---- |
>   | ResNet50 [r1,r2]  |    41.5    |   37.4   |
>   | DeiT-S |     46.4      |   39.9  |
>   | TNT-S      |     48.1      |  41.5  |
>
>   [r1] Lin, Tsung-Yi, et al. "Feature pyramid networks for object detection." CVPR 2017.
>   [r2] Chen, Kai, et al. "MMDetection: Open mmlab detection toolbox and benchmark." arXiv preprint arXiv:1906.07155 (2019).
>
> **Q4**: Since TNT specifically focuses on ... iNaturalist19 can be provided to further evaluate the effectiveness of the proposed method.
>
> **A4**: Thanks for the nice suggestion. We evaluate TNT-B on the large-scale fine-grained dataset iNaturalist19. From the results, we can see that TNT-B can obtain 1.8\% accuracy gain over DeiT-B, which indicates the effectiveness for fine-grained visual understanding tasks. We'll include this in the paper.
>
>   | Backbone | \#params (M) | Acc  |
>   | -------- | ------------ | ---- |
>   | DeiT-B   |     86.4         |   81.4   |
>   | TNT-B    |     65.6         |   83.2   |

---

> > ### Comment · Reviewer_6Dax · 2021-08-18
> > **Post-rebuttal comment**
> >
> > The authors' responses partially resolve my concerns.
> >
> > Specifically, A1 provides experimental results to validate that the performance gain originates from better pixel-level representation, instead of simply improved model depth. A2 provides a more systematic comparison with the DeiT baseline, which proves that the proposed inner transformer block can bring consistent improvement. A4 shows TNT is competitive on the large-scale fine-grained dataset. I'm satisfied with these responses and convinced by the rich experimental results.
> >
> > However, A3 does not completely resolve my concern on TNT's ability to adapt to the object detection task, which significantly benefits from multi-scale and high-resolution features. Though high AP can be achieved, the attention mechanism inevitably brings high memory consumption on high-resolution images, which can be unacceptable for many popular GPU devices. A similar concern is also raised by reviewer u1uh in A3.

---

> > > ### Author Response · Authors · 2021-08-19
> > > **Authors' response**
> > >
> > > Thanks for your support. How to reduce the memory useage on high-resolution images caused by attention mechanism while maintaining high performance is an interesting topic. And we will continue to address this issue in the future work.

---

### Official Review · Reviewer_Gg7D · 2021-07-16

**Rating:** 6
**Confidence:** 5

**Summary:**

The work proposes a nested vision Transformer variant (TNT), which adds an additional inner Transformer to capture within-patch information. By setting the hidden dimension of the inner Transformer much smaller, the FLOPs and #Params of the proposed TNT won't be overly large. Empirically, authors show that the proposed model performs better than ViT/DeiT on ImageNet classification, COCO object detection and ADE20K semantic segmentation.

**Limitations And Societal Impact:**

The propose method should bring in any additional limitation of negative societal impact when compared to ViT.

**Main Review:**

The idea of using a nested Transformer for visual processing is intuitive and reasonable. From the perspective of FLOPs and #Params, the complexity analysis "partially" address the concern of why adding another Transformer which processes the input at a much fine-grained level won't be too expensive.

However, since the inner Transformer has to process the image at the "word" level, the number FFN projections is dominated by `n x m` (`n` is the number of patches and `m` is the number of sub-patches in each patch), which is much larger than the number of FFN projections in ViT. While using a small inner hidden dim (`c`) seems to resolve the problem in complexity, I was wondering how much this actually slows down the running time (training & inference), since many smaller matrix multiplications is usually slower than few larger matrix multiplications of the same FLOPs under modern hardwares (e.g. GPU with tensor cores and TPU). This information is not mentioned or discussed in the current version.

Another question I have is whether one needs such a complicated inner network (Transformer) to capture the local granularity (16x16 pixels), especially if we are concerned with the ultimate efficiency. Experimenting with some simpler architectures may give more insights for this question.

**Time Spent Reviewing:**

2

---

> ### Author Response · Authors · 2021-08-09
> **Authors' response**
>
> We thank the reviewer for detailed comments.
>
> **Q1**: However, since the inner Transformer has to ... This information is not mentioned or discussed in the current version.
>
> **A1**: Thanks for the nice suggestion.  We'll discuss the CPU/GPU latency of the proposed TNT architecture in the updated version.
>
> First, the inference speed on CPU is satisfied as shown in the table. The latency is measured on an Intel CPU with batch size of 1.
> Compared with the baseline DeiT, the proposed TNT can achieve a better latency/acc trade-off.
>
> | Model  | FLOPs (B) | CPU latency (s) | ImageNet Top-1 |
> | ------ | --------- | --------------- | ----- |
> | TNT-S  | 5.2       | 5.9             | 81.5  |
> | TNT-B  | 14.1      | 9.7             | 82.8  |
> | DeiT-S | 4.6       | 4.7             | 79.8  |
> | DeiT-B | 17.6      | 9.9             | 81.8  |
>
> Second, since we introduce some additional subtle operations and calculations, which cannot well utilize the computing power of GPU. Thus, the GPU latency of TNT is worse than that of DeiT. However, the training/inference speed on GPU can be further optimized in the engineering level or model level.
>
> 1. The current implementation is based on PyTorch which is not optimal for speed. To achieve higher speed on GPU/TPU, more engineering works like CUDA implementation or hardware unit support, are required to handle imbalanced matrix multiplication. For the common settings of DeiT-S/TNT-S, $n=196$, $d=384$, $m=16$ and $c=24$. In the outer transformer, the typical matrix multiplication is $\mathbb{R}^{196\times 384} \times \mathbb{R}^{384\times 384}$, with FLOPs of 28,901,376. While in the inner transformer, the typical matrix multiplication is $\mathbb{R}^{(196*16)\times 24} \times \mathbb{R}^{24\times 24}$, with FLOPs of 1,806,336. So the shape of matrix in the inner transformer is extremely imbalanced and we will continue to optimize in our future work.
> 2. We can optimize the speed in the model level. Since the resolution and content inside the patch is smaller than that of the whole image, we may need fewer blocks to learn its representation. Thus, we can reduce the used TNT blocks and replace some with vanilla transformer blocks. As shown in the table, we can achieve a better accuracy-speed trade-off by using fewer TNT blocks. For example, TNT-S-1 with 4 TNT blocks can achieve 81.4 ImageNet Top-1 with the inference speed of 668 images/s. The accuracy-speed trade-off is shown in this figure (https://i.postimg.cc/RhCbKZ88/GPU-latency.png).
>
> | Model           | Indices of TNT blocks        | FLOPs (B) | Training speed (images/s, 8\*V100) | Inference speed (images/s, V100) | ImageNet Top-1 |
> | --------------- | ---------------------------- | --------- | --------------------------------- | -------------------------------- | ----- |
> | TNT-S           | [1,2,3,4,5,6,7,8,9,10,11,12] | 5.2       | ~1000                              | 428                              | 81.5  |
> | TNT-S-1         | [1,4,8,12]                   | 4.8       | ~1900                              | 668                              | 81.4  |
> | TNT-S-2         | [1,6,12]                     | 4.7       | ~2100                              | 704                              | 81.3  |
> | TNT-S-3         | [1,6]                        | 4.7       | ~2400                              | 757                              | 81.1  |
> | TNT-S-4         | [1]                          | 4.6       | ~2900                              | 822                              | 80.8  |
> | TNT-S-5         | [6]                          | 4.6       | ~2900                              | 822                              | 80.8  |
> | TNT-S-6         | [12]                         | 4.6       | ~2900                              | 822                              | 80.6  |
> | DeiT-S          | -                            | 4.6       | ~3200                              | 907                              | 79.8  |
> | PVT-Small [r1]  | -                            | 3.8       | -                                 | 820                              | 79.8  |
> | PVT-Medium [r1] | -                            | 6.7       | -                                 | 526                              | 81.2  |
> | DeiT-B          | -                            | 17.6      | ~1700                              | 292                              | 81.8  |
>
> [r1] Wang, Wenhai, et al. "Pyramid vision transformer: A versatile backbone for dense prediction without convolutions." ICCV 2021.
>
> **Q2**: Another question I have is whether one needs such a ... may give more insights for this question.
>
> **A2**: Thanks for the nice concern. We evaluate simpler architectures: only MSA or MLP. From the results in the following table, we can see that inner network with only MSA or MLP performs worse than the vanilla inner transformer. Both components are necessary since MSA can build the relationship between sub-patches, while MLP can perform space transformation and nonlinearity for each sub-patch.
>
> | Model   | Architecture of inner network | \#params (M) | FLOPs (B) | ImageNet Top-1 |
> | ------- | ----------------------------- | ------------ | --------- | ----- |
> | TNT-S   | Transformer block             | 23.8         | 5.2       | 81.5  |
> | TNT-S-1 | Transformer block w/o MSA     | 23.8         | 5.1       | 80.4  |
> | TNT-S-2 | Transformer block w/o MLP     | 23.8         | 5.1       | 80.8  |

---

> > ### Author Response · Authors · 2021-08-18
> > **Fix figure link**
> >
> > Thanks for the addtional question from Reviewer u1uh. The x-axis in the old figure (https://i.postimg.cc/RhCbKZ88/GPU-latency.png) is speed (images/s) instead of latency (s). The fixed figure is availble at https://i.postimg.cc/XNwr7TqD/GPU-speed.png.

---

### Official Review · Reviewer_xZPU · 2021-07-17

**Rating:** 7
**Confidence:** 4

**Summary:**

This paper presents a new Transformer-based framework "Transformer-iN-Transformer" (TNT) for visual recognition. The proposed TNT module can better preserve and model the local information for visual recognition compared to conventional VIT. Extensional experiments prove its effectiveness in image classification and object detection.

**Main Review:**

Originality:

This idea is novel, especially because the proposed framework is a leading work that views each patch as a sequence of sub-patches (visual words).

---

Quality:

The quality of this paper is good in general. Everything looks fair and reasonable.

---

Clarity:

The paper is well-written and easy to follow. The figures and tables in this paper are neat and clear.

---

Significance:

The paper will be impactful as it is one of the leading works that explore the locality of vision Transformer.

---

Overall, I agree with the contributions claimed in this paper and I suggest this paper be accepted.


---

Summary: This paper presents a new Transformer-based framework "Transformer-iN-Transformer" (TNT) for visual recognition. The proposed TNT module can better preserve and model the local information for visual recognition compared to conventional VIT. Extensional experiments prove its effectiveness in image classification and object detection.

**Time Spent Reviewing:**

1

---

> ### Author Response · Authors · 2021-08-09
> **Authors' response**
>
> Thanks for your strong support and detailed comments.

---

### Decision · Program_Chairs · 2021-09-27

**Decision:**

Accept (Poster)

**Comment:**

In this paper, the authors propose a simple but effective ViT network architecture, which introduces hierarchical structure in transformers. The architecture is motivated well and the paper is well-organized.

Although in the submission, the empirical study is not sufficient to the advantage of the proposed model, during the rebuttal period, the authors did a great job to provide more evidences. It is a clear acceptance for this paper. Please include all the additional empirical experiments results into the final version, especially the running time in training & inference stage and the necessity of inner transformer.